# Development and Validation of a New Method for Detecting Acetic Bacteria in Wine

**DOI:** 10.3390/foods12203734

**Published:** 2023-10-11

**Authors:** Alejandro Parra, Aroa Ovejas, Lucía González-Arenzana, Ana Rosa Gutiérrez, Isabel López-Alfaro

**Affiliations:** 1Laboratorio Dolmar Tentamus, Paraje Micalanda, 26221 Gimileo, La Rioja, Spain; alejandro@dolmarlaboratorio.com (A.P.); aroa@dolmarlaboratorio.com (A.O.); 2ICVV, Instituto de Ciencias de la Vid y del Vino (Universidad de La Rioja, Gobierno de La Rioja and CSIC), Finca La Grajera, Ctra. Burgos km 6, 26007 Logroño, La Rioja, Spain; lucia.gonzalez@unirioja.es (L.G.-A.); ana-rosa.gutierrez@unirioja.es (A.R.G.)

**Keywords:** acetic acid, acetic bacteria, olfactometry, wine

## Abstract

In winemaking, excessive production of acetic acid by acetic acid bacteria poses a major challenge, leading to rejection of wine by consumers. The aim of this study was to devise an economically viable and easy-to-use liquid culture medium for the preventive detection of microorganisms capable of generating acetic acid in wine. The modified medium incorporated specific nutrients that favored the growth of acetic acid bacteria and increased selectivity. Under varying conditions and with different types of wine, this medium was tested together with inoculated samples, comparing the occurrence of acetic acid and olfaction. The result was a new liquid medium based on olfactometry, designed to facilitate its use in wineries, even by untrained personnel and without the need for complex laboratory equipment. Validation was carried out on a variety of wines, determining the onset of the presence of acetic acid in the medium. This innovative culture medium provides a means to estimate the concentration of micro-organisms capable of producing acetic acid in wine. Its application in wineries facilitates proactive decision making, avoiding undesirable increases in acetic acid concentration.

## 1. Introduction

The presence of acetic bacteria in wine can generate acetic acid in concentrations above the perception threshold, causing the wine to be rejected by the consumer. The threshold levels of acetic acid for a wine to be considered organoleptically unsuitable vary between 0.4 and 1.5 g/L, depending on the type of wine and the consumer [1].

During the winemaking process it is very difficult to avoid the presence of acetic bacteria, as there is always a residual population that accompanies the wine [2]. Because of this, after certain treatments involving aeration (pumping, racking), small amounts of acetic acid can be produced. But the problem occurs when these microorganisms develop in large quantities and large amounts of this acid are generated. Traditionally, this situation has been described in poorly preserved wines, especially as a result of contact with air [3].

To avoid the content of acetic acid in wine increasing, it is necessary to control the factors that favor the development of acetic bacteria, such as the presence of residual sugars and especially oxygen, which is the limiting factor in the growth of these microorganisms. The absence of oxygen does not eliminate bacteria, but it does prevent their growth and slow down their metabolism. Other factors are pH and temperature. The optimum pH for the growth of acetic bacteria is between 5 and 6, so high pH favors their development in wine [4]. The optimum growth temperature of acetic bacteria is around 30 °C, although much slower growth of acetic bacteria has been detected even at 10 °C [5]. Other key points to prevent the development of these microorganisms are hygiene in the facilities and the elements in which the wine is stored. The bottling process is one of the critical points, since oxygen can remain in the head space of the bottle and also through the sealing material of the containers. Even by correctly controlling all critical points, acetic bacteria are not eliminated from the wine, so their development is only blocked, preventing acetic acid from forming. To eliminate them, an amicrobic filtration of the wine should be made, a technique that is not usual at the winery level and that is often undesirable because it detracts from the sensory quality of the wine.

The ability of acetic bacteria to present an adaptive cell state known as Viable But Not Culturable (VBNC) bacteria allows their survival for long periods of time under unfavorable conditions. When acetic bacteria are in the VBNC state, they are not detected by “traditional” tests and can cause “acetic piqure” in wines that have been previously tested and are considered not to contain acetic bacteria [6].

Although acetic bacteria are the main producers of acetic acid in wine, other oenological microorganisms can also produce this acid. Saccharomyces cerevisiae yeast generates small amounts of acetic acid [7] during alcoholic fermentation, but it is non-Saccharomyces yeasts (*Brettanomyces bruxellensis* [8], *Hanseniaspora guilliermondii*, *Kloeckera apiculata* [9]) that are the main producers. Heterofermentative lactic acid bacteria (*Oenococcus oeni, Pediococcus* spp., *Lactobacillus* spp.) present in wine also slightly increase volatile acidity during malolactic fermentation and during wine preservation under the right conditions [10]. Acetogenic bacteria (*Acetobacterium* and *Clostridium* [11]) are also capable of producing it, but are not common in wine. The new molecular biology techniques applied to enology have made it possible to expand the knowledge of the microbiota of wines and their relationship with metabolic processes. This has meant that the number of known microorganisms capable of generating acetic acid is increasing, but acetic bacteria remain the main cause of acetic piqure [12,13,14,15].

There are several techniques to determine the presence of acetic bacteria in wine. The most widely used in the winery are usually solid culture media. Several solid culture media are available for the analysis of acetic bacteria, but in enology those described by the International Organization of Vine and Wine (OIV) are mainly used. In OIV/OENO resolution 206/2010 (Microbiological analysis of wines and musts. Detection, differentiation and counting of microorganisms), the use of three means for the counting of *Acetobacteria* is suggested: GYC, G2 and Kneifel. Another frequently used medium is the Wallerstein medium [16]. Other more precise techniques, such as PCR and flow cytometry, are being implemented slowly and less widely, due to their cost and the fact that they require qualified technical personnel. 

There has been growing concern among wineries to control the presence of acetic bacteria and the increase in the concentration of acetic acid in wines during their conservation. This is because there are no simple and effective analyses to allow the early detection of these microorganisms. The solid culture media available for the detection of acetic bacteria in wine do not always work well, due to their low sensitivity. The use of the quantitative real-time PCR (qPCR) technique provides very good results in the detection and quantification of acetic bacteria [17], but most wineries cannot perform this type of analysis because they do not have the equipment required in their laboratories and because outsourcing the analysis is quite expensive. This fact makes it necessary to develop simple, effective and inexpensive methods that allow an early detection of acetic bacteria in wineries.

The objective of this study was to create a new and user-friendly method for detecting acetic bacteria in wineries. This approach enhances sensitivity compared to existing methods, enabling timely decision-making by accurately predicting the population of these microorganisms in wine. To do this, we relied on the work carried out by Couto et al. (2005) [18] and Rodrigues et al. (2001) [19] in which they designed a liquid culture medium for the early detection of Brettanomyces yeasts based on olfactometry.

## 2. Materials and Methods

### 2.1. Culture Media

The culture medium used for this study was developed from the culture media recommended by the OIV for the analysis of acetic bacteria in wine (GYC, G2, Kneiffel) [20] and the Wallerstein medium [16]. Modifications were introduced to these media, both in their composition and in the incubation conditions. 

### 2.2. Acetic Acid Analysis

Acetic acid was determined by enzymatic analysis in accordance with OIV/OENO resolution 391/2010. An automatic MIURA-200 sequential analyzer from TDI Tecnología Difusión Ibérica S.L. (Barcelona, Spain) was used. 

### 2.3. Quantitative PCR

Commercial Scorpions kits from ETS Laboratories (St. Helena, CA, USA) have been used to carry out DNA extraction, and specific probes for each group of microorganisms analyzed in the wine. 

Sample preparation: The samples were centrifuged at 3500 rpm for 10 min, the supernatant was removed, and 1 mL of the washing solution was added, which was then recentrifuged. 

Cellular Lysis: 100 µL of enzyme mix were added, homogenized and incubated at 37 °C for 40 min, stirring every 10 min. Afterwards, 120 µL of proteinase K solution, which contains an internal control, was homogenized, and incubated at 60 °C for 20 min. Then, 100 µL of lysis solution were added, mixed and incubated at 70 °C for 10 min. This was centrifuged at 6000 rpm for 15 s, and the supernatant was collected. 

ADN Extraction: 200 µL of ethanol were added to the supernatant, mixed and centrifuged at 12,000 rpm for 1 min in a collector tube. The tube was then washed with solutions available in the kit. The tube was dried and the DNA was eluted by adding 50 µL of elution solution at 70 °C, leaving for 1 min at room temperature, and the process was then repeated, centrifuged for 1 min at 12,500 rpm, and the extract collected. 

Preparation of reactions: In each PCR tube, 10 µL of reaction medium, 5 µL of probes, 4.7 µL of water, 0.3 µL of internal pattern probes and 5 µL of DNA extract were added.

Amplification program: denaturation at 95 °C for 300 s; amplification for 40 cycles (93 °C–10 s, 60 °C–35 s); fluorophore used for acetic bacteria is FAM (λ 495/520 nm). The thermal cycler used was Cepheid’s SmartCycler (Sunnyvale, CA, USA). 

In the qPCR analysis of the wines, one group of microorganisms was studied, grouped according to the primers available in the Scorpions kit, the group of acetic bacteria *Acetobacter* spp. 

### 2.4. Microorganisms

The culture medium developed was validated by inoculation tests with different microorganisms, both individually and with mixtures thereof. The strains of microorganisms were obtained from the CECT, Colección Española de Cultivos Tipo (Universitat de València, Paterna, Valencia). The strains used were as follows: *Acetobacter aceti* (CECT 298), *Saccharomyces cerevisae* (CECT 1182), *Pichia membranafaciens* (CECT 1115), *Gluconobacter oxidans* (CECT 4009), *Acetobacter oeni* (CECT 5830), *Komagataeibacter europaeus* (CECT 7583), *Penicillium* spp. (CECT 20566), *Brettanomyces* spp. (CECT 14517) and *Oenococcus oeni* (CECT 217).

The instructions provided by the CECT for the recovery of freeze-dried cultures were followed. Once reconstituted, the strains were diluted 104 times in peptone water (15 g/L of peptone, 10 g/L of tryptone and 5 g/L of sodium chloride) and the diluted solution was sown in the PCA medium (2.5 g/L Yeast extract, 1 g/L Glucose, 5 g/L Enzyme Digest of Casein, 15 g/L Bacteriological Agar). They were incubated at 25 °C for 48 h. Subsequently, the concentration of viable microorganisms was calculated by counting colonies in the plate. These strains were used to seed the culture medium at different concentration levels.

### 2.5. Olfaction Tests

The detection of the presence of acetic acid by olfaction was carried out daily in the jars with the culture medium seeded with different microorganisms or wines. The test was carried out by a tasting panel formed by six tasters. This tasting panel is accredited in ISO 17025 [21] by the National Accreditation Entity in Spain (ENAC) of Laboratorios Dolmar Tentamus. Each day the tasters noted the presence or absence of acetic acid in the smell of the sample. 

### 2.6. Commercial Wines

Both in the tests carried out with the available culture media and in the development and validation tests of the newly designed medium, commercial wines naturally contaminated with microorganisms were used. In total, 80 wines were analyzed, including the 25 used in the development of the medium, from different Spanish regions (Bierzo, Cadiz, Campo de Borja, Cariñena, Castilla-La Mancha, Extremadura, Madrid, Navarra, Priorat, Rias Baixas, Ribeiro, Ribera del Duero, Rioja, Rueda, Sevilla, Somontano, Toro and Valencia) and different grape varieties (Albariño, Bobal, Cabernet Sauvignon, Garnacha red and white, Godello, Malbec, Merlot, Palomino, Pedro Ximenez, Syrah, Tempranillo red and white, Teixadura, Verdejo and Viura). 

All samples were analyzed by qPCR before inoculating them in the different culture media. 

## 3. Results

### 3.1. Development of the Culture Medium

To start the design of a new culture medium, the efficacy of the three culture media suggested by the OIV for the detection of acetic bacteria in wine (GYC, G2, Kneiffel) and the Wallerstein medium (WLL) was first tested. For the comparative study of the media with respect to their detection sensitivity, 25 commercial wines were used. These wines were also used in the following stages throughout the design and conservation process of the culture medium. The results obtained by seeding the wine samples in the different culture media are shown in Table 1. This table also indicates the number of acetic bacteria determined by qPCR. As can be seen, the wines contained variable populations between <10 and 7.9 × 10^5^ cell/mL.

As can be seen in Table 1, the results obtained when the wines were seeded in the culture media recommended by the OIV (GYC, G2, Kneiffel) were mostly negative, and growth was only observed in the plates in two samples out of the 25 analyzed, which corresponded to populations of acetic bacteria greater than 10^5^ cell/mL. The composition of these culture media is very different in terms of glucose concentration (50 g/L, 0, 0, respectively), yeast extract (10, 1.2, 30 g/L, respectively), or presence of other compounds such as calcium carbonate or ethanol, but none of them was suitable for detecting the presence of acetic bacteria at low concentration. The exception was the Wallerstein medium, in which growth was observed in 18 out of the 25 wines, when they had populations greater than 9.6 × 10^2^ cell/mL. The composition of this medium is more complex in terms of micronutrients, which would explain its better detection. However, in this case it was not possible to detect populations of acetic bacteria smaller than 10^3^ cell/mL. These low values can compromise the evolution of the wine and cause deterioration during conservation. These results demonstrate that the determination of the population of acetic bacteria in traditional culture media can be improved, due to its low sensitivity. 

To do this, starting from the Wallerstein medium in liquid form, without agar and without bromocresol green, variations were made in its composition to increase its sensitivity. Two media were initially prepared by modifying the amount of trace elements to promote the growth of microorganisms. When evaluating their sensitivity, interferences of their components in the aroma were detected, which hindered the olfactory detection of acetic acid. Therefore, another two media were designed, in which yeast extract and peptone were replaced by pea extract and tryptose, to try to minimize the initial aroma of the medium. In these latter two media, the addition of 2% of amberlite in water was also tested. The initial aromas of the media were not very different, and only the addition of amberlite reduced the initial aromas of the media, making it easier to detect the acetic acid aroma later on. However, these changes did not significantly increase the detection levels by the tasting panel. Therefore, it was decided not to use the amberlite treatment because it made the medium preparation process more difficult and increased the final price of the product. The culture medium that was finally considered more suitable had a similar but simplified composition to the Wallerstein medium, without KCl and FeCl_3_ and with the addition of bile salts and pimaricin. 

The medium finally chosen was distributed in 50 mL plastic pots previously sterilized by immersion in ethanol. These pots were kept at room temperature for one week and their evolution during this time of conservation was studied. It was found that after one week, 50% of the pots had become cloudy and developed mold growth. Therefore, tests were subsequently carried out on the viability, preservation and stability of the medium, because the aim was for the medium to have as long a shelf life as possible and to be easy to store. To achieve this, 100 mg/L of natamycin (pimaricin) was added to inhibit yeast growth and 12.5 mg/L of penicillin to prevent the growth of lactic bacteria. Violet crystal (5 μL) and bile salts (0.5 g/L) were added to the culture medium before sterilizing, as they are inhibitors of Gram + bacteria; and after sterilizing 0.04 g/L cycloheximide to inhibit fungal growth and 66 mg/L pimaricin were also added to the medium. The final composition of the medium is described in Table 2. Yeast extract, peptone, glucose, CaCl_2_, MgSO_4_, MnSO_4_ and H_2_PO_4_, were used, supplemented with water, and adding pimaricin, cycloheximide, bile salts and violet crystal to inhibit interference. The incubation conditions were 30 °C for 14 days.

Next, the stability of the medium thus prepared was checked under different conditions: refrigeration at 4 °C, at room temperature and at 30 °C, and the appearance of alterations in the culture medium was verified weekly. The most suitable storage conditions were storage in a refrigerated chamber at 4 °C, since after 6 months 100% of the pots remained stable.

Once the composition of the medium was established, it was prepared for use in validation tests. The medium was prepared in glass jars and sterilized in autoclave (121 °C for 15 min). Subsequently, 20 mL of medium were distributed in plastic pots of 50 mL capacity in a laminar flow chamber with ultraviolet light after prior sterilization by immersion for 24 h in a 70% ethanol solution. The pots were stored in a chamber at 4 °C before use. 

The conditions established for the correct use of the culture medium were as follows: 20 mL of the sample to be analyzed was added to the pot with the culture medium and incubated in an oven at 30 °C. Every 24 h, the pots were analyzed by olfaction and the presence of acetic acid aromas was noted. The test was performed daily until the detection was positive or for up to a maximum of 14 days. 

### 3.2. Validation of the Culture Medium

The validation of the medium was carried out by seeding it with microorganisms of collection, both individually and combined, and with wines contaminated in a natural or artificial way. Once the medium was seeded with the samples, aliquots were taken every one or two days of incubation to analyze the evolution of the concentration of acetic acid. At the same time, the medium was controlled by olfaction to determine the moment at which the threshold of perception of acetic acid was exceeded. The verification of the ability of acetic bacteria to grow in the culture medium was determined by analyzing this medium by q-PCR before and after incubation in some samples.

#### 3.2.1. Growth in the Medium of Acetic Bacteria

Figure 1 represents the evolution of the concentration of acetic acid over time in the medium inoculated with different concentrations of a pure culture of *Acetobacter aceti*. The concentration of acetic acid was observed to increase over the days, and the faster the rate, the higher the initial concentration of bacteria. In this way, the initial concentration of acetic bacteria could be estimated from the moment at which the threshold of perception of acetic acid (0.7 g/L) is exceeded [22].

But during winemaking, other genera of acetic bacteria, besides *Acetobacter*, can contaminate wines and increase volatile acidity. Figure 2 and Figure 3 show the variation in acetic acid concentration over time when the culture medium is inoculated with three genera of acetic bacteria (*Acetobacter*, *Gluconobacter or Komagaitebacter*) at different levels. When the initial inoculated population was low (10^3^ CFU/mL), the growth of *Gluconobacter* and *Komagaitebacter* was slower (Figure 2). However, when the initial population was higher (10^7^ CFU/mL), the increase in acetic acid concentration was very rapid and the kinetics were similar among the three bacteria studied. This confirms the suitability of the culture medium for the detection of acetic acid produced by different genera of acetic bacteria in a range of 5 to 10 days depending on the initial population and the genus of acetic bacteria present in the wine.

#### 3.2.2. Growth in the Medium of Other Oenological Microorganisms

The rate of acetic acid production in the culture medium over time (Table 3) of different oenological microorganisms was compared. In addition to the four species of acetic bacteria (*Acetobacter aceti*, *Acetobacter oeni*, *Gluconobacter oxidans*, *Komagataeibacter europaeus*), the lactic acid bacteria responsible for malolactic fermentation in wines (*Oenococus oeni*), and the main yeast responsible for alcoholic fermentation (*Saccharomyces cerevisae*) were studied. Both microorganisms remain in the wine for months and are capable of producing acetic acid, although if the winemaking conditions are controlled, its production is small [7]. However, during the preservation of finished wines, other microorganisms, in addition to acetic bacteria, can develop and increase the volatile acidity of the wine. This is the case of the yeasts *Pichia membranafaciens* and *Brettanomyces* spp. [8]. To assess the possible interference of these microorganisms, with the contamination produced by acetic bacteria, the medium was inoculated at a rate of 10^4^ CFU/mL of these microorganisms and their production of acetic acid was compared with that produced by acetic bacteria under the same incubation conditions. The results shown in Table 3 show that the microorganisms studied did not present a relevant acetic acid production during the first 7 days, possibly due to their inability to grow in the culture medium. However, at this time, all acetic bacteria produced high amounts of acetic acid, greater than 1 g/L, clearly detectable by olfaction. The table indicates in bold the acetic acid value and the day of incubation on which acetic acid odor was clearly detected in the samples. As can be seen, the olfactory detection was only positive in the case of acetic bacteria, at 5, 6 and 7 days depending on the species. This result would indicate that the presence of such micro-organisms would not be confused with the presence of acetic acid bacteria. Thus, they would not interfere with the analysis and would not lead to errors in the estimation of acetic acid bacteria in the samples.

However, during storage, different types of micro-organisms coexist with acetic acid bacteria in finished wines. These micro-organisms may interact with each other and cause an increase in acetic acid, which may not be exclusively attributable to acetic acid bacteria. To evaluate these possible interferences between microorganisms that could cause errors in detection, the culture medium was inoculated together with different acetic bacteria (10^4^ CFU/mL), and with each of the five microorganisms studied above (10^4^ CFU/mL). The results shown in Table 4 indicate that during the first 7 days after seeding the medium, the rate of acetic acid accumulation is determined by the presence of acetic bacteria, and that the presence of the other microorganisms does not modify the olfactory detection time. As shown in Table 4, the combined inoculation of different microorganisms with *Acetobacter aceti* was positive in olfactometry on the fifth day of incubation, and at 6 and 7 days in the case of the other two species of acetic bacteria. These are the same times as observed when acetic acid bacteria were inoculated alone (Table 3).

#### 3.2.3. Verification of the Usefulness of the Medium in Commercial Wines

To study the usefulness of the culture medium and olfactory detection, it was seeded with 80 commercial wines containing different microbial loads depending on their origin, age, etc. The wines were analyzed by qPCR before adding them to the culture medium, to know the population of acetic bacteria they contained. Once the culture medium was inoculated with the different wines, the concentration of acetic acid and its aroma were monitored daily by olfaction. Table 5 shows the results obtained, and as can be seen, the initial concentration of acetic bacteria could be related to the time it takes to perceive acetic acid aroma in the samples. The olfactometric test was positive at 4 days of incubation when the presence of acetic bacteria in the wines was of the order of 10^5^–10^4^ cell/mL, 5 days for levels of 10^4^–10^3^, 6 days for levels of 10^3^–10^2^, and 7–8 days for levels of 10^2^–10^1^. For contents of acetic bacteria less than 10 cell/mL the test would be negative for 14 days. 

Taking into account these data, a guide has been designed for the interpretation of the results obtained when using the culture medium with commercial samples. This guide (Table 6) was designed on the basis of a table for the yeast Brettanomyces designed by C. Gerland, (Intelli’Oeno, Bourg-Lès-Valence, France), which has not been published. The guide indicates the estimated population of acetic acid bacteria present based on the time it takes for the acetic acid odor to develop, and also, which treatments or precautions should be applied depending on the result obtained. Along with Table 6, instructions on how to use the culture medium in several steps are indicated. First, 10 mL of wine is added to the bottle of culture medium. The bottle is placed in a stove at 30 °C, and every 2 days the bottle is removed and the appearance of acetic acid is checked by olfaction. 

## 4. Conclusions

In conclusion, we have successfully developed a new refrigerated and stable liquid culture medium that offers sensitivity and selectivity for the detection and semi-quantification of acetic acid bacteria in wines by olfaction. Our study demonstrated a strong correlation between the results obtained using this medium and those derived from qPCR analysis, validating its reliability. This innovative method is not only cost-effective, but also very easy to use, as it only requires basic equipment such as a cooker and a refrigerator. In addition, minimal training is sufficient for people to detect the aroma associated with the presence of acetic acid in the medium. Looking ahead, this analysis promises to be a proactive tool to prevent unexpected spoilage related to acetic acid bacteria during wine storage. To assist in this endeavor, we have developed a practical guide outlining strategies to mitigate the risk of acetic acid formation based on the findings of our analysis. However, it is essential to recognize that, like any study, our work has its limitations, and that future research may explore new, broader applications and possible refinements.

## Figures and Tables

**Figure 1 foods-12-03734-f001:**
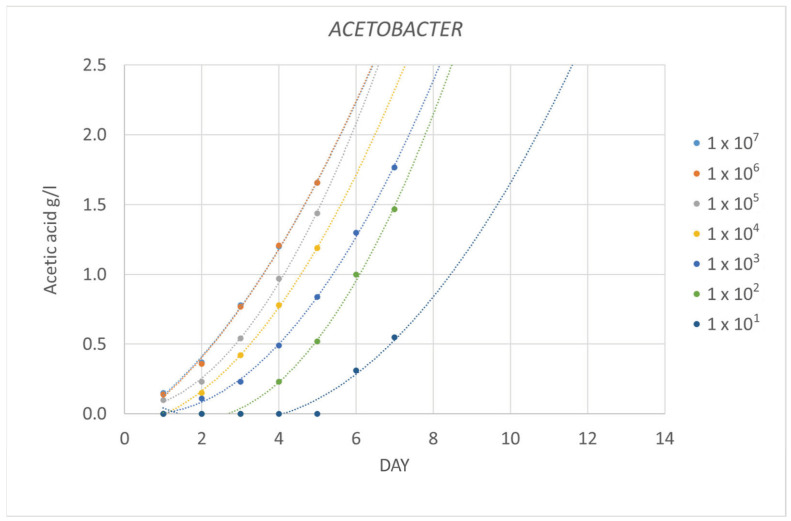
Evolution of the average concentration of acetic acid (*n* = 3) in a medium inoculated with different populations of *Acetobacter aceti* (CFU/mL).

**Figure 2 foods-12-03734-f002:**
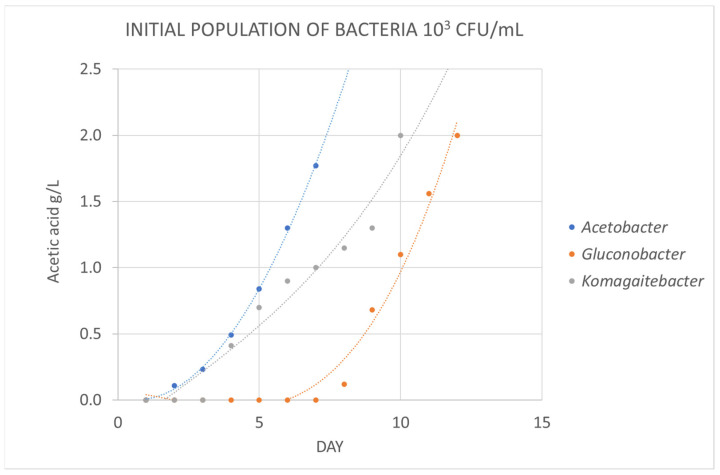
Evolution of the average concentration of acetic acid (*n* = 3) in a medium inoculated with acetic bacteria in a population 10^3^ CFU/mL.

**Figure 3 foods-12-03734-f003:**
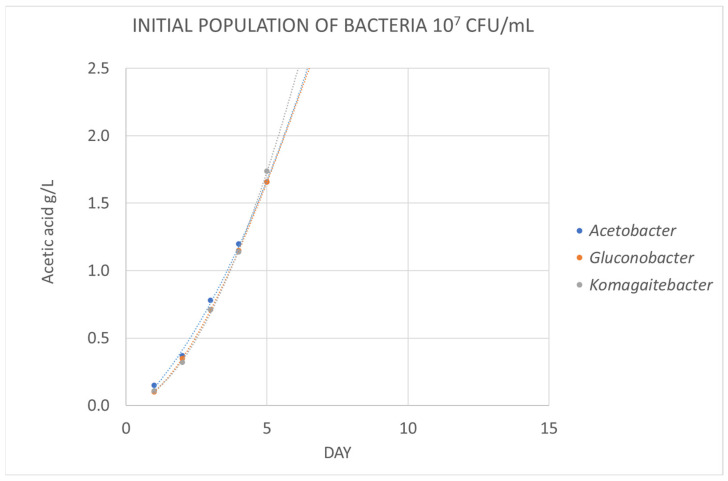
Evolution of the average concentration of acetic acid (*n* = 3) in a medium inoculated with acetic bacteria in a population 10^7^ CFU/mL.

**Table 1 foods-12-03734-t001:** Population of acetic bacteria present in 25 wines, analyzed by q-PCR and presence/absence of growth in different culture media.

Sample	Region	Variety and/or Type	Cell/mL Acetic Bacteria	Medium GYC	Medium G2	Medium Kneifel	Medium Wallerstein
WINE 1	Rioja	Tempranillo Red	2.4 × 10^4^	No	No	No	Yes
WINE 2	Rioja	Garnacha Red	2.7 × 10^3^	No	No	No	Yes
WINE 3	Rioja	Graciano Red	3.7 × 10^4^	No	No	No	Yes
WINE 4	Rioja	Tempranillo White	1.2 × 10^1^	No	No	No	No
WINE 5	Rioja	Viura White	4.8 × 10^2^	No	No	No	No
WINE 6	Ribera Duero	Tempranillo Red	7.9 × 10^5^	Yes	Yes	Yes	Yes
WINE 7	Ribera Duero	Tempranillo Red	4.1 × 10^4^	No	No	No	Yes
WINE 8	Ribera Duero	Tempranillo Red	6.9 × 10^3^	No	No	No	Yes
WINE 9	Ribera Duero	Tempranillo Red	1.4 × 10^3^	No	No	No	Yes
WINE 10	Rueda	Verdejo White	<10	No	No	No	No
WINE 11	Valencia	Bobal Red	5.1 × 10^4^	No	No	Yes	Yes
WINE 12	Valencia	Merlot Red	7.1 × 10^3^	No	No	No	Yes
WINE 13	Extemadura	Syrah Red	5.5 × 10^4^	Yes	No	No	Yes
WINE 14	Extramadura	Malbec Red	1.8 × 10^3^	No	No	No	No
WINE 15	Cadiz	Palomino White	9.6 × 10^2^	No	No	No	Yes
WINE 16	Cadiz	Palomino White	5.4 × 10^3^	No	No	No	Yes
WINE 17	Cadiz	Pedro Ximénez White	4.8 × 10^3^	No	No	No	Yes
WINE 18	Priorat	Garnacha Red	3.6 × 10^4^	No	No	No	Yes
WINE 19	Priorat	Cabernet Sauvignon Red	3.9 × 10^5^	Yes	Yes	Yes	Yes
WINE 20	Navarra	Garnacha Rosé	4.6 × 10^3^	No	No	No	Yes
WINE 21	Navarra	Garnacha Rosé	8.1 × 10^3^	No	No	No	Yes
WINE 22	Rias Baixas	Albariño	2.4 × 10^2^	No	No	No	No
WINE 23	Rias Baixas	Treixadura	3.1 × 10^3^	No	No	No	No
WINE 24	Rias Baixas	Albariño	3.5 × 10^3^	No	No	No	Yes
WINE 25	Rias Baixas	Godello	9.1 × 10^1^	No	No	No	No

**Table 2 foods-12-03734-t002:** Final liquid culture medium for semi-quantification of acetic bacteria.

Components	Quantity
Yeast extract	4 g
Peptone	5 g
Glucose	50 g
H_2_PO_4_	0.55 g
CaCl_2_	0.125 g
MgSO_4_ × 7H_2_O	0.125 g
MnSO_4_	0.0025 g
H_2_O	Complete up to 1000 mL
Add before sterilizing 0.5 g of bile salts and 5 microliters of violet crystal. Add after sterilization 0.3 mL of natamycin (pimaricin) (concentration of 66 mg/20 mL) and 0.04 g of cycloheximide.	

**Table 3 foods-12-03734-t003:** Average concentration (g/L) of acetic acid (uncertainty of 6%) (*n* = 3) over 10^4^ CFU/mL. * Shading is marked when the aroma is detected by sniffing.

Microorganism	Day 1	Day 2	Day 3	Day 4	Day 5	Day 6	Day 7	Day 8	Day 9	Day 14
*Oenococus oeni*	0	0	0	0	0	0	0	0.05	0.1	0.15
*Pichia membranafaciens*	0	0	0	0	0	0	0.08	0.12	0.12	0.18
*Saccharomyces cerevisae*	0	0	0	0	0	0	0	0.08	0.10	0.15
*Brettanomyces* spp.	0	0	0	0	0	0	0.10	0.12	0.12	0.18
*Penicillium* spp.	0	0	0	0	0	0	0	0.10	0.10	0.15
*Acetobacter aceti*	0	0.15	0.42	0.78	* 1.19					
*Acetobacter oeni*	0	0.1	0.38	0.77	1.15					
*Gluconobacter oxidans*	0	0	0	0.19	0.37	0.68	1.05			
*Komagataeibacter europaeus*	0	0	0	0.53	0.86	1.19	1.49			

**Table 4 foods-12-03734-t004:** Average concentration (g/L) of acetic acid (uncertainty of 6%) (*n* = 3) over time in samples contaminated with different microorganisms in conjunction with acetic bacteria. Shading is marked when the aroma is detected by sniffing.

Combined Microorganisms	Day 1	Day 2	Day 3	Day 4	Day 5	Day 6	Day 7
*Oenococus oeni*, *Acetobacter aceti*	0	0.10	0.38	0.69	1.51		
*Oenococus oeni*, *Acetobacter oeni*	0	0.12	0.37	0.66	1.45		
*Oenococus oeni*, *Gluconobacter oxidans*	0	0	0	0.15	0.34	0.62	0.98
*Oenococus oeni*, *Komagataeibacter europaeus*	0	0	0	0.46	0.78	1.09	
*Pichia membranafaciens*, *Acetobacter aceti*	0	0.08	0.36	0.65	1.47		
*Pichia membranafaciens*, *Acetobacter oeni*	0	0.10	0.39	0.67	1.56		
*Pichia membranafaciens*, *Gluconobacter oxidans*	0	0	0	0.14	0.36	0.65	1.01
*Pichia membranafaciens*, *Komagataeibacter europaeus*	0	0	0	0.39	0.74	1.01	
*Saccharomyces cerevisae*, *Acetobacter aceti*	0	0.10	0.39	0.71	1.64		
*Saccharomyces cerevisae*, *Acetobacter oeni*	0	0.11	0.36	0.67	1.67		
*Saccharomyces cerevisae*, *Gluconobacter oxidans*	0	0	0	0.21	0.40	0.65	1.07
*Saccharomyces cerevisae*, *Komagataeibacter europaeus*	0	0	0.10	0.41	0.76	1.05	
*Brettanomyces* spp., *Acetobacter aceti*	0	0.11	0.40	0.70	1.35		
*Brettanomyces* spp., *Acetobacter oeni*	0	0.10	0.38	0.67	1.42		
*Brettanomyces* spp., *Gluconobacter oxidans*	0	0	0	0.17	0.35	0.60	0.99
*Brettanomyces* spp., *Komagataeibacter europaeus*	0	0	0	0.29	0.70	0.99	
*Penicillium* spp., *Acetobacter aceti*	0	0.14	0.49	0.72	1.41		
*Penicillium* spp., *Acetobacter oeni*	0	0.11	0.45	0.68	1.46		
*Penicillium* spp., *Gluconobacter oxidans*	0	0	0	0.14	0.31	0.69	0.98
*Penicillium* spp., *Komagataeibacter europaeus*	0	0	0	0.47	0.78	1.14	

**Table 5 foods-12-03734-t005:** Results of acetic acid and olfaction obtained with commercial wines of different origins and varieties, with the culture medium designed.

Wine	Origin	Type	qPCR *Acetobacter* Cell/mL	Initial Acetic Acid g/L	Positive Day	Final Acetic Acid g/L	Olfaction
26	Rioja	Tempranillo Red	1.8 × 10^5^	0.37	4	0.99	YES
1	Rioja	Tempranillo Red	2.4 × 10^4^	0.26	4	0.73	YES
6	Ribera Duero	Tempranillo Red	7.9 × 10^5^	0.30	4	0.81	YES
7	Ribera Duero	Tempranillo Red	4.1 × 10^4^	0.27	4	0.72	YES
11	Valencia	Bobal Red	5.1 × 10^4^	0.24	4	0.74	YES
18	Priorat	Garnacha Red	3.6 × 10^4^	0.25	4	0.74	YES
19	Priorat	Cabernet Sauvignon Red	3.9 × 10^5^	0.24	4	0.82	YES
23	Rias Baixas	Treixadura	3.1 × 10^3^	0.12	4	0.75	YES
48	Extremadura	Red	2.4 × 10^5^	0.4	4	0.79	YES
61	Navarra	Garnacha Rosé	6.8 × 10^5^	0.28	4	1.16	YES
67	Rías Baixas	White	1.9 × 10^5^	0.35	4	0.81	YES
78	La Mancha	White Sweet	1.5 × 10^5^	0.31	4	1.20	YES
2	Rioja	Garnacha Red	2.7 × 10^3^	0.19	5	1.02	YES
3	Rioja	Graciano Red	3.7 × 10^4^	0.22	5	1.05	YES
8	Ribera Duero	Tempranillo Red	6.9 × 10^3^	0.17	5	1.01	YES
9	Ribera Duero	Tempranillo Red	1.4 × 10^3^	0.19	5	0.95	YES
12	Valencia	Merlot Red	7.1 × 10^3^	0.17	5	1.02	YES
13	Extemadura	Syrah Red	5.5 × 10^4^	0.21	5	1.08	YES
14	Extramadura	Malbec Red	1.8 × 10^3^	0.21	5	0.96	YES
16	Cadiz	Palomino White	5.4 × 10^3^	0.13	5	0.91	YES
17	Cadiz	Pedro Ximénez White	4.8 × 10^3^	0.16	5	0.97	YES
20	Navarra	Garnacha Rosé	4.6 × 10^3^	0.18	5	1.02	YES
21	Navarra	Garnacha Rosé	8.1 × 10^3^	0.20	5	0.91	YES
24	Rias Baixas	Albariño	3.5 × 10^3^	0.15	5	0.94	YES
40	Rioja	Red	6.9 × 10^4^	0.18	5	0.96	YES
46	Rioja	Rosé	1.6 × 10^4^	0,2	5	0.89	YES
57	Ribera Duero	Red	9.9 × 10^4^	0.18	5	0.97	YES
62	Navarra	Rosé	7.1 × 10^4^	0.21	5	0.93	YES
63	Madrid	Red	4.2 × 10^4^	0.25	5	1.25	YES
68	Rías Baixas	White	6.7 × 10^4^	0.15	5	0.96	YES
76	La Mancha	Red	8.0 × 10^4^	0.23	5	1.07	YES
80	Yecla	Red	3.6 × 10^4^	0.2	5	0.77	YES
5	Rioja	Viura White	4.8 × 10^2^	0.16	6	1.05	YES
15	Cadiz	Palomino White	9.6 × 10^2^	0.12	6	0.96	YES
22	Rias Baixas	Albariño	2.4 × 10^2^	0.13	6	0.95	YES
41	Rioja	Red	4.3 × 10^4^	0.19	6	1.13	YES
44	Rioja	White	3.8 × 10^3^	0.1	6	0.86	YES
45	Rioja	Rosé	1.3 × 10^4^	0.24	6	0.95	YES
47	Rioja	Rosé	2.3 × 10^3^	0.17	6	0.91	YES
49	Ribera Duero	Red	7.0 × 10^3^	0.32	6	0.88	YES
51	Ribera Duero	Red	2.5 × 10^3^	0.32	6	0.95	YES
53	Ribera Duero	Red	3.5 × 10^3^	0.35	6	1.14	YES
69	Rías Baixas	White	3.1 × 10^3^	0.21	6	0.83	YES
70	Rías Baixas	White	3.7 × 10^3^	0.22	6	0.93	YES
75	La Mancha	Red	7.8 × 10^3^	0.2	6	0.94	YES
43	Rioja	Red	4.5 × 10^2^	0.11	7	0.95	YES
52	Ribera Duero	Red	4.3 × 10^2^	0.2	7	0.83	YES
54	Rioja	Red	1.4 × 10^3^	0.16	7	0.91	YES
56	Toro	Red	1.7 × 10^2^	0.21	7	0.93	YES
58	Navarra	Red	2.7 × 10^2^	0.13	7	0.89	YES
59	Navarra	Rosé	3.5 × 10^2^	0.15	7	1.08	YES
72	Ribeiro	White	2.8 × 10^2^	0.15	7	0.84	YES
77	La Mancha	White	6.9 × 10^2^	0.14	7	1.02	YES
4	Rioja	Tempranillo White	1.2 × 10^1^	0.08	8	0.89	YES
25	Rias Baixas	Godello	9.1 × 10^1^	0.09	8	0.93	YES
50	Ribera Duero	VRed	5.7 × 10^2^	0.17	8	0.98	YES
71	Rías Baixas	White	1.2 × 10^1^	0.14	11	0.98	YES
73	Ribeiro	Red	3.6 × 10^1^	0.15	11	0.94	YES
10	Rueda	Verdejo White	<10	0.06	14	0.12	NO
27	Rioja	Tempranillo Red	<10	0.23	14	1.01	YES
28	Rias Baixas	Albariño White	<10	0.09	14	0.17	NO
29	Rias Baixas	Albariño White	<10	0.14	14	0.21	NO
30	Rias Baixas	Albariño White	<10	0.14	14	0.23	NO
31	Rioja	Tempranillo White	<10	0.06	14	0.10	NO
32	Cariñena	Garnacha Red	<10	0.12	14	0.41	NO
33	Rias Baixas	Albariño White	<10	0.08	14	0.15	NO
34	Ribeiro	Red	<10	0.16	14	0.27	NO
35	Ribeiro	Red	<10	0.16	14	0.26	NO
36	Cadiz	Oloroso	<10	0.42	14	0.44	NO
37	Sevilla	White	<10	0.13	14	0.15	NO
38	Campo Borja	Rosé semisweet	<10	0.12	14	0.18	NO
39	Rioja	Red	<10	0.21	14	0.28	NO
42	Rioja	Red	<10	0.24	14	0.28	NO
55	Hungría	White	<10	0.1	14	0.20	NO
60	Navarra	Rosé	<10	0.08	14	0.21	NO
64	Priorat	Red	<10	0.06	14	0.13	NO
65	Priorat	Red	<10	0.12	14	0.17	NO
66	Rías Baixas	White	<10	0.06	14	0.21	NO
74	Bierzo	Red	<10	0.07	14	0.21	NO
79	Jerez	White Sweet	<10	0.08	14	0.18	NO

**Table 6 foods-12-03734-t006:** Table for the interpretation of the results obtained with the culture medium developed.

Days Necessary for the Appearance of the Odour	Estimated Population of Acetic Bacteria	What to Do?
>12 days/No appearance	Absence in 20 mL	Control in 1 month
10 days	Very weak (about 100 bacteria/mL)	Control in 15 days
8 days	Weak (100 to 1000 bacteria/mL)	Control in 1 week
6–7 days	Media (1000 to 10,000 bacteria/mL)	2 controls: immediate and after 5 days
4 days	Significant: Danger (10,000 to 100,000 bacteria/mL)	ACT: Filtration/Centrifugation /Flash-pasteurization Add SO_2_ After a few days return to control
3 days	Strong: A lot of danger (+1,000,000 bacteria/mL)	

## Data Availability

The data underlying this article are available in the article.

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
