# Peer review of "Development and Validation of a New Method for Detecting Acetic Bacteria in Wine"

_foods, 2023, doi:10.3390/foods12203734_

Round 1

Reviewer 1 Report

Comments and Suggestions for Authors

Dear Editor,

The original and novel manuscript entitled “Development and validation of a new method for detecting acetic bacteria in wine” is appropriately well written, developed and structured by Parra et al. in suitable English with a clear structure. They developed and evaluated a liquid culture medium for identification and detection of acetic acid producing microorganisms. This study is very interesting and practical. There are some major concerns and minor revisions which should be clarified and addressed.

Major concerns

-       Each methodology study which has developed a new diagnostic or identification procedure should evaluate the technical sensitivity (accuracy), technical specificity, experimental sensitivity, experimental specificity and reproducibility of the new method. I found that the authors evaluated the sensitivity of the new method but they did not explain that how they did it. Also, there are no results regarding the other parameters that I mentioned. The authors need to explain how they evaluate the sensitivity and also evaluate other parameters of the new method including technical specificity, experimental sensitivity, experimental specificity and reproducibility. Also, they should explain these evaluations in the methods section of the manuscript.   

-       It is necessary to clarify that what are the positive and negative controls. Please tabulate all negative and positive acetic acid organisms and explain that how you used them in this study. There are limited control (also negative controls) organisms in this study. Please explain that why did you use this limited number of control strains. How do you prove the validity of the test by this limited number of negative controls?

Minor revisions

-       You need to add an introduction sentence as the first sentence of the abstract section.

-       Please add the details information regarding the primers you used if it is possible.

-       Please add all figures regarding the qPCR method as the supplementary data in your study.

Author Response

Major concerns

Each methodology study which has developed a new diagnostic or identification procedure should evaluate the technical sensitivity (accuracy), technical specificity, experimental sensitivity, experimental specificity and reproducibility of the new method. I found that the authors evaluated the sensitivity of the new method but they did not explain that how they did it. Also, there are no results regarding the other parameters that I mentioned. The authors need to explain how they evaluate the sensitivity and also evaluate other parameters of the new method including technical specificity, experimental sensitivity, experimental specificity and reproducibility. Also, they should explain these evaluations in the methods section of the manuscript.

The sensitivity of the new method has been evaluated by inoculating different microorganisms in the new culture medium, and controlling the acetic acid concentration. Acetic acid has been controlled by olfaction and by enzymatic analysis, in the latter case the sensitivity, specificity and reproducibility of the method is accredited by ISO 17025. Regarding the specificity of the medium, it has been evaluated in table 3, which shows different characteristic microorganisms in oenology, whether or not they are capable of producing acetic acid, and in combination with each other, table 4, it has been proven whether they interfere.

Table 5 demonstrates the sensitivity of the method by studying 79 real wine samples, in which the acetic acid bacteria content was analysed by PCR, and it was found that what has already been studied by inoculation of bacteria in the medium is reproducible in the wines, and indicates on the day of appearance of the acetic aroma, how sensitive the medium is.

It is necessary to clarify that what are the positive and negative controls. Please tabulate all negative and positive acetic acid organisms and explain that how you used them in this study.There are limited control (also negative controls) organisms in this study. Please explain that why did you use this limited number of control strains. How do you prove the validity of the test by this limited number of negative controls?

The validity of the test is shown in table 5, wines with no acetic acid bacteria show no increase in acetic acid after 2 weeks, and those with acetic acid bacteria show an increase in acetic acid that is faster the higher the concentration of acetic acid bacteria.

The microorganisms that have been used have been selected for their relevance in oenology, and it must also be taken into account that wine is a limiting medium.

The culture medium developed is intended to be used in wineries, so it has been studied with the microorganisms that are mostly found in wines. We do not believe that it is appropriate to do a greater number of negative controls, since the idea is that we are able to detect if a wine is going to have problems due to the appearance of acetic acid.

Minor revisions

You need to add an introduction sentence as the first sentence of the abstract section.

It has been corrected in the article.

Please add the details information regarding the primers you used if it is possible.

The primers are from a commercial company, with copyright, and are not public.

Please add all figures regarding the qPCR method as the supplementary data in your study.

I don't quite understand what you mean. There are many samples, 79 wines, as well as medium and other inoculated wines.

We attach PCR screenshots of a day in which some wines were analyzed.

Reviewer 2 Report

Comments and Suggestions for Authors

Dear Editor and Authors,

The authors of the article titled: Development and validation of a new method for the detection of acetic acid bacteria in wine, aimed to develop an economically viable, easy-to-use liquid culture medium for the preventive detection of the presence of microorganisms capable of producing acetic acid in wine. The article is poorly written and presented in a relatively simplistic way, and its aims and of course the whole study look very interesting, but unfortunately the article also contains very incomprehensible things, some methods are not well described. The authors always worked in three replicates, but the article lacks any statistics (there is only an arithmetic average). The article is not badly written, but many things need to be corrected and improved. A few comments that need to be incorporated into the article before it is further processed:

- Write all Latin names of microorganisms in italics.

- do not write spp. in italics! correct throughout the document

- standardise the units - it is more appropriate to use g/L - correct throughout the document

- L183, 189, 193 are missing L - cell/ml

- L278 missing superscript

- Figures 1, 2 and 3, Table 3 - all entries n=3 must be in italics!

- Table 3 has a mistake in the title - fix it... over Table 104 - there are extra periods and brackets!

- Table 5 - do not use commas between numbers! Use dots! Scan and fix.

- Table 6 - they use a different font.

Overall: I recommend the author to rewrite the conclusion and focus especially on the results obtained, future perspectives and limitations of this study! I also miss the novelty of this study! I recommend putting it under the introduction section and highlighting it. I also recommend rewriting the abstract, which should mainly reflect the aims and the most important, less achieved results!

Comments on the Quality of English Language

Moderate editing of English language is required.

Author Response

The authors of the article titled: Development and validation of a new method for the detection of acetic acid bacteria in wine, aimed to develop and economically viable, easy-to-use liquid culture medium for the preventive detection of the presence of microorganisms capable of producing acetic acid in wine. The article is poorly written and presented in a relatively simplistic way, and its aims and of course the whole study look very interesting, but unfortunately the article also contains very incomprehensible things, some methods are not well described. The authors always worked in three replicates, but the article lacks any statistics (there is only an arithmetic average). The article is not badly written, but many things need to be corrected and improved. A few comments that need to be incorporated into the article before it is further processed:

Please, we do not understand very well what you mean with the description of the methods. In the case of the concentration of acetic acid, the tables indicate that the results have an associated uncertainty of 6%, which is the one calculated in the accreditation of this method in ISO 17025. The method developed is olfactory, and is qualitative, I'm not sure what statistic you are referring to, the smell is in days.

- Write all Latin names of microorganisms in italics.

- do not write spp. in italics! correct throughout the document

- standardise the units - it is more appropriate to use g/L – correct throughout the document

- L183, 189, 193 are missing L - cell/ml

- L278 missing superscript

- Figures 1, 2 and 3, Table 3 - all entries n=3 must be in italics!

- Table 3 has a mistake in the title - fix it... over Table 104 - there are extraperiods and brackets!

- Table 5 - do not use commas between numbers! Use dots! Scan and fix.

- Table 6 - they use a different font.

It has been corrected in the article.

Overall: I recommend the author to rewrite the conclusion and focus especially on the results obtained, future perspectives and limitations of this study! I also miss the novelty of this study! I recommend putting it under the introduction section and highlighting it. I also recommend rewriting the abstract, which should mainly reflect the aims and the most important, less achieved results!

It has been corrected in the article.

Reviewer 3 Report

Comments and Suggestions for Authors

The paper presents a novel culture medium for the detection of acetic bacteria in wine. The authors have tested the proposed culture medium against reference culture media and have shown how the proposed medium can detect acetic bacteria at lower concentrations. Wine samples of different origins and different microorganisms have been tested. I think the following revisions are needed to improve the paper quality.

1) At the end of the introduction there is the sentence “The aim of this study was to develop a method for detecting acetic bacteria that is easy to use by wineries. To do this, we relied on the work carried out by Couto et al. (2005) [18] and Rodrigues et al. (2001) [19] in which they designed a liquid culture medium for the early detection of Brettanomyces yeasts based on olfactometry and that favored the development of viable but non-cultivable bacteria [20]”. Please, explain how the proposed method improves the methods discussed in the above references.

2) I think a paragraph should be added where the step-by-step procedure to detect acetic bacteria in wine using the proposed culture medium is presented.

3) At lines 280-282 there is the sentence “The rate of acetic acid production in the culture medium over time (Table 3) of different oenological microorganisms was compared. In addition to the 4 species of acetic bacteria (Acetobacter aceti, Acetobacter oeni, Gluconobacter oxidans, Komagataeibacter europaeus)”. However, data for Acetobacter oeni are not present in Table 3.

4) The authors should check the manuscript for errors and typos. For example, in the caption of Figure 3 and Table 3, exponents should be superscript.

Comments on the Quality of English Language

The English is generally good with only minor errors and typos to fix as discussed in the comments to the authors.

Author Response

The paper presents a novel culture medium for the detection of acetic bacteria in wine. The authors have tested the proposed culture médium against reference culture media and have shown how the proposed medium can detect acetic bacteria at lower concentrations. Wine simples of different origins and different microorganisms have been tested. I think the following revisions are needed to improve the paper quality.

1) At the end of the introduction there is the sentence “The aim of this study was to develop a method for detecting acetic bacteria that is easy to use by wineries. To do this, we relied on the work carried out by Couto et al.(2005) [18] and Rodrigues et al. (2001) [19] in which they designed a liquid culture medium for the early detection of Brettanomyces yeasts based on olfactometry and that favored the development of viable but non-cultivable bacteria [20]”. Please, explain how the proposed method improves the methods discussed in the above references.

The aforementioned articles describe the use of a culture medium for the yeast Brettanomyces and how its concentration in wine is related to the appearance of the 4-ethylphenol molecule. In our case, a similar methodology has been followed, based on these articles but for acetic bacteria. The use of a simple medium that allows a characteristic odour to be related to the concentration of a microorganism is an idea that appears in these articles, and on which we are based, but the final objective is totally different.

2) I think a paragraph should be added where the step-by-step procedure to detect acetic bacteria in wine using the proposed culture medium is presented.

It has been corrected in the article and highlighted in the article, line 341:

Along with Table 6, instructions on how to use the culture medium in several steps are indicated. First, 10 ml of wine is added to the bottle of culture medium. The bottle is placed in a stove at 30 ºC, and every 2 days the bottle is removed and the appearance of acetic acid is checked by olfaction.

3) At lines 280-282 there is the sentence “The rate of acetic acid production in the culture medium over time (Table 3) of different oenological microorganisms was compared. In addition to the 4 species of acetic bacteria (Acetobacter aceti, Acetobacter oeni, Gluconobacter oxidans, Komagataeibacter europaeus)”. However, data for Acetobacter oeni are not present in Table 3.

Sorry, it was a mistake. It has been corrected in the article.

4) The authors should check the manuscript for errors and typos. For example, in the caption of Figure 3 and Table 3, exponents should be superscript.

It has been corrected in the article.

Round 2

Reviewer 1 Report

Comments and Suggestions for Authors

Dear Authors,

Please provide point-by-point responses to my comments. I have only received a file that includes some real-time PCR software photos. Also, some of my comments have not been addressed in the manuscript. I am looking forward to receiving the revised version of your manuscript. If my comments are not addressed by next time, I will have to reject your manuscript. 

Author Response

There must have been a mistake.  I entered the answers to your comments in the box and also sent them to the editor in an attachment. I am going to attach a file with the comments, I hope that now they will reach you. I send you my apologies.

Reviewer 2 Report

Comments and Suggestions for Authors

Dear Editor and Authors,

In my opinion, the authors have improved their manuscript. As for the statistics, I thought that if you work in 3 replicates, then I am missing at least the standard deviation in the tables (minimum and basic statistics - i.e. the result +/- standard deviation). But the article makes a good impression on me and I think it is very interesting. The authors did a very good job with the addition and highlighting of the novelties of the article, which I very much appreciate. After some minor changes, I recommend publishing the article in journal Foods.

-Some names of microorganisms are not in italics (e.g. L132). Please check the whole document!

Comments on the Quality of English Language

Minor editing of English language is still required.

Author Response

I agree with your remarks regarding the statistical study, but on this occasion, it was considered a better option not to increase the size and complexity of the tables and to publish the data with only the uncertainty described in each of the tables.

-Some names of microorganisms are not in italics (e.g. L132). Please check the whole document!

It has been corrected in the article.

-Minor editing of English language is still required.

This manuscript has been edited twice for English language by an English professional editing service.

Reviewer 3 Report

Comments and Suggestions for Authors

The authors revised the paper according to the Reviewer comments. It can be accepted for publication.

Author Response

Thank you very much for your comments as well as your revision of the manuscript.

Round 3

Reviewer 1 Report

Comments and Suggestions for Authors

Dear Authors,

Thank you very much for your responses and I have no more comments. I recommend publishing this valuable manuscript.